# GILBO: One Metric to Measure Them All

**Alexander A. Alemi**,* **Ian Fischer**\*
Google AI
{alemi,iansf}@google.com

## Abstract

We propose a simple, tractable lower bound on the mutual information contained in the joint generative density of any latent variable generative model: the GILBO (*Generative Information Lower BOund*). It offers a data independent measure of the complexity of the learned latent variable description, giving the log of the effective description length. It is well-defined for both VAEs and GANs. We compute the GILBO for 800 GANs and VAEs trained on MNIST and discuss the results.

## 1 Introduction

GANs (Goodfellow et al., 2014) and VAEs (Kingma & Welling, 2014) are the most popular latent variable generative models, because of their relative ease of training and high expressivity. However *quantitative* comparisons across different algorithms and architectures remains a challenge. VAEs are generally measured using the ELBO, which measures their fit to data. Many metrics have been proposed for GANs, including the INCEPTION score (Gao et al., 2017), the FID score (Heusel et al., 2017), independent Wasserstein critics (Danihelka et al., 2017), birthday paradox testing (Arora & Zhang, 2017), and others.

Instead of focusing on metrics tied to the data distribution, we believe a useful additional independent metric worth consideration is the complexity of the trained generative model. Such a metric would help answer questions related to overfitting and memorization, and may also correlate well with sample quality. To work with both GANs and VAEs our metric should not require a tractable joint density $p(x, z)$. To address these desiderata, we propose the GILBO.

## 2 GILBO: Generative Information Lower BOund

A symmetric, non-negative, reparameterization independent measure of the information shared between two random variables is given by the mutual information:

$$I(X; Z) = \int dx\, dz\, p(x, z) \log \frac{p(x, z)}{p(x)p(z)} = \int dz\, p(z) \int dx\, p(x|z) \log \frac{p(x|z)}{p(x)} \geq 0. \quad (1)$$

$I(X; Z)$ measures how much information (in nats) is learned about one variable given the other. As such it is a measure of the complexity of the generative model. It can be interpreted (when converted to bits) as the reduction in the number of yes-no questions needed to guess $X$ if you observe $Z$, or vice-versa. It gives the log of the *effective description length* of the generative model. This is roughly the log of the number of distinct samples (Tishby & Zaslavsky, 2015). $I(X; Z)$ is well-defined even for continuous distributions. This contrasts with the continuous entropy $H(X)$ of the marginal distribution, which is not reparameterization independent (Marsh, 2013). It is intractable due to the presence of $p(x)$, but we can define a tractable variational lower bound (Agakov, 2006):

$$\text{GILBO} \equiv \int dz\, p(z) \int dx\, p(x|z) \log \frac{e(z|x)}{p(z)} = \mathbb{E}_{p(x,z)} \left[ \log \frac{e(z|x)}{p(z)} \right] \leq I(X; Z) \quad (2)$$

We call this bound the GILBO for *Generative Information Lower BOund*. It requires learning a tractable variational approximation to the intractable posterior $p(z|x) = p(x, z)/p(z)$, termed $e(z|x)$ since it acts as an *encoder* mapping from data to a prediction of its associated latent variables.[1]

---

*Authors contributed equally.

[1]Note that a *new* $e(z|x)$ is trained for both GANs and VAEs. VAEs do not use their own $e(z|x)$.

The encoder $e(z|x)$ performs a regression for the inverse of the GAN or VAE generative model, approximating the latents that gave rise to an observed sample. This encoder should be a tractable distribution, and must respect the domain of the latent variables, but does not need to be reparameterizable, as no sampling from $e(z|x)$ is needed during training. We suggest the use of $(-1, 1)$ remapped Beta distributions in the case of uniform latents, and Gaussians in the case of Gaussian latents. Optimizing the GILBO for the parameters of the encoder gives a lower bound on the true generative mutual information in the GAN or VAE.

The GILBO contrasts with the *representational mutual information* of VAEs defined by the data and encoder, which motivates VAE objectives (Alemi et al., 2017). For VAEs, both lower and upper variational bounds can be defined on the representational joint distribution $(p(x)e(z|x))$. These have demonstrated their utility for cross-model comparisons. However, they require a tractable posterior, preventing their use with most GANs. The GILBO finally provides a theoretically-justified and dataset-independent metric that allows direct comparison of VAEs and GANs.

The GILBO is entirely independent of the *true* data, being purely a function of the generative joint distribution. This makes it distinct from other proposed metrics like estimated marginal log likelihoods (often reported for VAEs and very expensive to estimate for GANs), an independent Wasserstein critic (Danihelka et al., 2017), or the common INCEPTION (Gao et al., 2017) and FID (Heusel et al., 2017) scores which attempt to measure how well the generated samples match the observed true data samples. Being independent of data the GILBO does not directly measure sample quality, but in practice it correlates well.

Even though the GILBO doesn't directly reference the dataset, the dataset provides useful signposts. First is at $\log C$, the number of distinguishable classes in the data. If the GILBO is lower than that, the model has almost certainly failed to learn a reasonable model of the data. Another is at $\log N$. A GILBO near this value may indicate that the model has largely memorized the training set, or that the model's capacity happens to be constrained near the size of the training set. At the other end is the entropy of the data itself ($H(X)$) taken either from a rough estimate, or from the best achieved data log likelihood of any known generative model on the data. Any reasonable generative model should have a GILBO no higher than this value.

Unlike other metrics, GILBO does not monotonically map to quality. Both extremes indicate failures. A vanishing GILBO denotes a generative model with vanishing complexity, either due to independence of the latents and samples, or a collapse to a small number of possible outputs. A diverging GILBO suggests oversensitivity to the latent variables.

## 3  EXPERIMENTS

We computed the GILBO for each of the 700 GANs and 100 VAEs tested in Lucic et al. (2017) on the MNIST dataset in their *wide range* hyperparameter search. This allows us to compare FID scores and GILBO scores for a large set of different GAN objectives on the same architecture. For our encoder network, we duplicated the discriminator, but adjusted the final output to be a linear layer predicting the $64 \times 2 = 128$ parameters defining a $(-1, 1)$ remapped Beta distribution (or Gaussian in the case of the VAE) over the latent space. We used a Beta since all of the GANs were trained with a $(-1, 1)$ 64-dimensional uniform distribution. The parameters of the encoder were optimized for up to 500k steps with ADAM (Kingma & Ba, 2015) using a scheduled multiplicative learning rate decay. We used the same batch size (64) as in the original training. Training time for estimating GILBO is comparable to doing FID evaluations (a few minutes).

In Figure 1 we show the distributions of FID and GILBO scores for all 800 models as well as their scatterplot. We can immediately see that each of the GAN objectives collapse to GILBO $\sim 0$ for some hyparameter settings, but none of the VAEs do. In Figure 2 we show generated samples from all of the models, split into relevant regions. A GILBO near zero signals a failure of the model to make any use of its latent space (Figure 2a). The best performing models by FID all sit at a GILBO $\sim 11$ nats. An MNIST model that simply memorized the training set and partitioned the latent space into 50,000 unique outputs would have a GILBO of $\log 50,000 = 10.8$ nats, so the cluster around 11 nats is suspicious. Among a large set of hyperparameters and across 7 different GAN objectives, we notice a conspicuous increase in FID score as GILBO moves away from $\sim 11$ nats to either side. This demonstrates a failure of existing GANs to achieve a meaningful range of complexities while

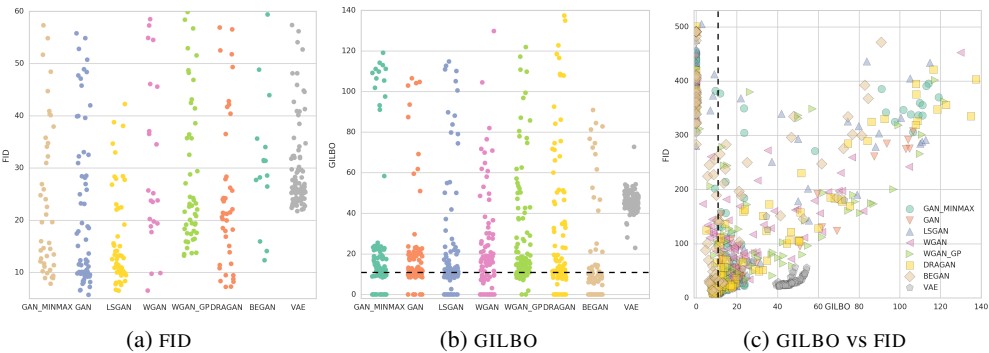

(a) FID  (b) GILBO  (c) GILBO vs FID

Figure 1: (a) is a recreation of Figure 5 (left) from Lucic et al. (2017) showing the distribution of FID scores for each model. Points are jittered to give a sense of density. (b) Shows the distribution of GILBO scores. (c) Shows FID vs GILBO.

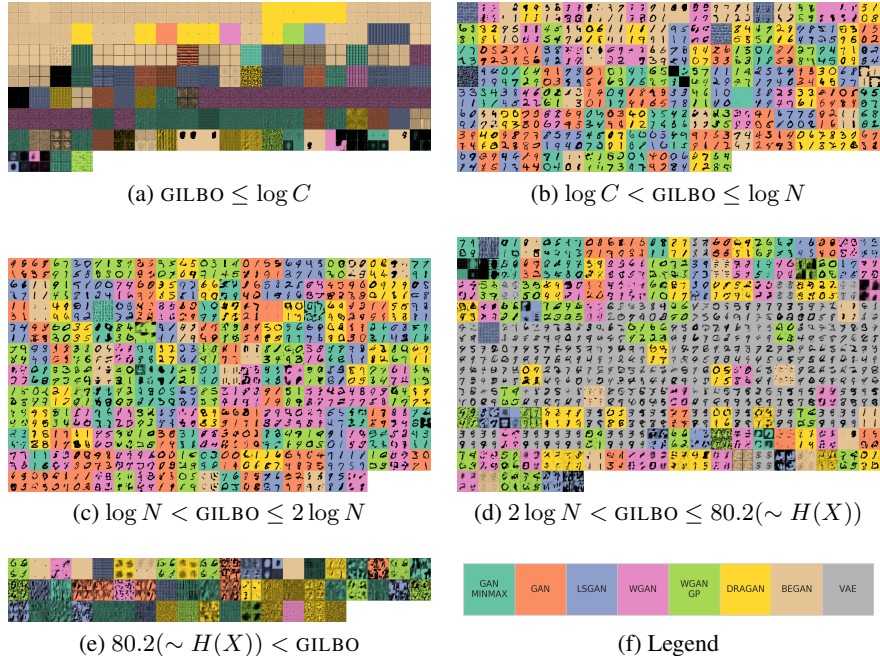

(a) GILBO $\leq \log C$  (b) $\log C <$ GILBO $\leq \log N$

(c) $\log N <$ GILBO $\leq 2 \log N$  (d) $2 \log N <$ GILBO $\leq 80.2 (\sim H(X))$

(e) $80.2 (\sim H(X)) <$ GILBO  (f) Legend

Figure 2: Samples from all models sorted by GILBO in raster order and broken up into representative ranges. The background colors correspond to the model family (Figure 2f). Note that all of the VAE samples are in (d), indicating that the VAEs achieved a non-trivial amount of complexity. Also note that most of the GANs in (d) have poor sample quality, further underscoring the apparent difficulty GANs have maintaining high visual quality without indications of training set memorization.

maintaining visual quality. Most striking is the distinct separation in GILBOs between GANs and VAEs. Clearly GANs learn less complex joint densities than a vanilla VAE.

## 4 CONCLUSION

We believe using GILBO for further comparisons across architectures, datasets, and GAN and VAE variants will illuminate the strengths and weaknesses of each. We think that GILBO should be reported when evaluating any latent variable model.

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
