# OpenReview forum: "GILBO: One Metric to Measure Them All"
_ICLR.cc/2018/Workshop — Accept_

### Official Review · AnonReviewer3 · 2018-03-08
**biased evaluation, unclear validity of metric**

**Rating:** 5
**Confidence:** 3

**Review:**

The authors propose to judge sample quality of generative models by approximating the mutual information between the latents z of the model, and the resulting image x. It's unclear why this would be a good metric. Ignoring problems with differential entropy on continuous variables, the mutual information between z and x would just be the entropy of z if the generative model decoder is deterministic and one to one. For generative models with this property (e.g. the NICE model) this means the proposed metric would only depend on the prior p(z) and would be completely unaltered by training the model.

In practice, the authors lower-bound the mutual information by training a probabilistic encoder. This results in a highly biased comparison between VAEs and GANs. VAEs are easier to invert (they have been trained to be) and therefore yield better lower bounds, while GANs are often impossible to invert since they are not one to one.

---

### Official Review · AnonReviewer2 · 2018-03-09
**Interesting way to evaluate GANs and VAEs**

**Rating:** 7
**Confidence:** 3

**Review:**


This work proposes using the lower-bound on the mutual information between the latent variable and the observed data to understand how much of information is captured in the latent variable. The authors propose using this as a data-independent proxy for the quality of a generative model. While the lower bound on the mutual information is itself not novel, the authors' proposal for its use as a quantitative metric appears new to me. Overall, the idea is interesting and the experiments study how a wide variety of GANS and VAEs (learned with different hyperparameters) behave as a function of the GILBO. The abstract is written clearly though it could use additional details and clarifications on how e(z|x) was learned and why one cannot use the learned inference network of a VAE (or in the case of a GAN: an adversarially learned inference network).

Pros:
* Interesting idea for a different way in which to evaluate GANs and VAEs, though it comes at the additional expense of learning e(z|x) for each model separately.
* Experiments: The authors give a breakdown of the range of values that the GILBO could take and then visualize the performance of the models along this axis.

Cons:
* One omission (in this work) that might prove interesting to study with the context of the GILBO is the InfoGAN -- since it explicitly a GAN learns while also maximizing the mutual information between the latent variable and observed data.
* The abstract is missing a discussion of https://openreview.net/pdf?id=B1M8JF9xx which uses AIS to form a quantitative metric to compare GANs and VAEs.

Question for the authors: How does a VAE with an overtly powerful decoder perform in terms of the GILBO?

---

### Official Review · AnonReviewer1 · 2018-03-09

**Rating:** 7
**Confidence:** 4

**Review:**

This paper proposes to use a lower bound on the mutual information between the joint density p_{gen}(x,z). The main idea is to train a parametric model to map from generated samples x to corresponding latent variables z.

The authors perform quite large number of experiments on GANs and VAEs trained on MNIST and compare their proposed score (GILBO) with other scores and do qualitative analysis of the sample quality

Overall I liked the proposed metric as it gives a new tool to compare the generative distributions of GANs and VAEs. I found the experiments on GANs quite interesting as they seem to suggest that GANs with high sample quality memorizes the data distribution. My main concerns are 1) how sensitive the bound is to the exact parameterisation and gradient descent training of e(z|x), 2) the experiments are only performed on MNIST and 3) that the lower bound is not a new contribution and  However, overall i think that the paper provides some nice contributions that is valuable to the ML community.


Q1: In general - How tight can we expect the bound on the mutual information to be ?
Q2: How sensitive is the lower bound on the exact parameterization of e(z |x)? Especially since we need to train this model using stochastic gradient descent it would be nice to see the variation for e(z|x) trained 10 times with different random seeds?


Q3: I’m not completely sure how the training works. In the abstract you state that you use a lower bound on “the joint generative density”. I interpret this as e(z|x) is fitted as e(z|x) = N(mu(x), sigma(x)), for normal density latent variable models, where x is samples from the generative distribution (x \sim p_generative) and z is the corresponding latent variables ? At least for GANs that seems like the only possible approach since we only know the mapping from z -> x. I think the paper could need some clarification here

Some other comments;:
I don’t think you support the following statement with experimental results - so that should be deleted i think.
“Being independent of data the GILBO does not directly measure sample quality, but in practice it correlates well. “
Also, please make the axis-labels on the figures a bit bigger.

---

### Decision · Program_Chairs · 2018-03-20
**ICLR 2018 Workshop Acceptance Decision**

**Decision:**

Accept

**Comment:**

Congratulations, your paper was accepted to the ICLR workshop. Given the concerns of reviewer 3, you may wish to soften the claims of generality, or the extent to which the measure can be compared between model classes.